# Barriers and Supports in eHealth Implementation among People with Chronic Cardiovascular Ailments: Integrative Review

**DOI:** 10.3390/ijerph19148296

**Published:** 2022-07-07

**Authors:** Sophia Herrera, Alide Salazar, Gabriela Nazar

**Affiliations:** 1Student Master’s Program in Nursing, Faculty of Nursing, Universidad de Concepción, Concepción 40730386, Chile; sophiaherrera@udec.cl; 2Faculty of Nursing, Universidad de Concepción, Concepción 40730386, Chile; 3Faculty of Social Sciences, Universidad de Concepción, Concepción 40730386, Chile; gnazar@udec.cl

**Keywords:** eHealth, chronic diseases, cardiovascular, interventions

## Abstract

eHealth interventions use information technology to provide attention to patients with chronic cardiovascular conditions, thereby supporting their self-management abilities. Objective: Identify barriers and aids to the implementation of eHealth interventions in people with chronic cardiovascular conditions from the perspectives of users, health professionals and institutions. Method: An integrative database review of WoS, Scopus, PubMed and Scielo of publications between 2016 and 2020 reporting eHealth interventions in people with chronic cardiovascular diseases. Keywords used were eHealth and chronic disease. Following inclusion and exclusion criteria application, 14 articles were identified. Results: Barriers and aids were identified from the viewpoints of users, health professionals and health institutions. Some notable barriers include users’ age and low technological literacy, perceived depersonalization in attention, limitations in technology access and usability, and associated costs. Aids included digital education and support from significant others. Conclusions: eHealth interventions are an alternative with wide potentiality for chronic disease management; however, their implementation must be actively managed.

## 1. Introduction

Chronic cardiovascular ailments are currently the main cause of general mortality [1]. One of the basic pillars of healthcare for patients with chronic conditions is their involvement in managing their own health, to promote and fortify better self-management [2]. In this environment, there is an emerging health strategy that uses information technology as a means for supporting users’ self-management. These are eHealth interventions involving the use of applications or web portals through smartphones, tablets or computers to send clinical data to a health team [3] or portals with educational content supporting patients at home. The European Communities Commission (ECC) has recently recognized the benefits of this type of intervention in managing chronic ailments, in terms of improving health-related quality of life (HRQL), medical attention quality and resource use, as well as its potential role for improving adherence to treatments, backing its adoption among member states [4].

Among the factors that could influence the effectiveness of eHealth interventions and, ultimately, their effective implementation are a series of elements tied to health organizations, health professionals and users [5] whose experience must be analyzed in order to understand the implementation process of this strategy, as well as to strengthen its benefits. Carrying out eHealth interventions with all these aspects in mind would improve their results regarding patient adherence and self-management [5], allowing for users’ active participation in their health, increasing their feeling of security and empowering them with regard to their chronic ailment [6]. Implementing an eHealth strategy in optimal conditions would definitely contribute to compensating people with chronic ailments, improving their quality of care, assigning resources more efficiently, improving cost efficiency, and diminishing complications and even mortality rates, as well as decreasing hospitalization and re-entry rates, along with greater satisfaction between patients and healthcare providers [7].

However, studies centered on barriers and aids in the user context are rare and, among them, even fewer address users’ subjective experiences [2]. Evidence in this area is insufficient, especially considering that most of these studies have been conducted in European and Asian countries. Some studies have described the lack of digital literacy as one of the main barriers, as well as lack of interest in and perceived utility of the implementation of eHealth interventions. The most common facilitators are family support and the constant accompaniment of the health team [8,9].

In this context, analyzing eHealth interventions would allow nursing professionals to not only judge their effectiveness, but also, by considering the experience of people with chronic ailments, become able to better understand how patients experience implementation, find out what barriers and aids were most relevant for their participation and what could have directly or indirectly influenced their effectiveness. With this information, health professionals could work together with users on diminishing barriers and maintaining aids in order to carry out better implementations on future projects and foreseeing these aspects.

Interventions in e-health are expected to continue to increase, as they are a cost-effective option, in the face of different limitations of traditional interventions, such as mobility restrictions, for example, those derived from the COVID-19 pandemic. Thus, knowing the experiences in the implementation of eHealth initiatives will help in their future development. This study addressed chronic cardiovascular diseases due to their high prevalence worldwide. A review of eHealth interventions is necessary due to the rapid increase in knowledge on the subject; moreover, reviews in recent years have focused on interventions aimed at specific chronic conditions, such as allergies [10] or depression [11], or included a variety of chronic conditions such as diabetes, HIV, pain, and epilepsy, among others [12]. Other reviews have focused exclusively on child or adolescent groups [11], or the older adult population [13]. Reviews of reviews on the subject were published more than 10 years ago [14].

The objective of this integrative review was to identify barriers and aids to eHealth intervention implementation in people with chronic cardiovascular conditions. The research question guiding the review was: What are the barriers and aids to implementing eHealth interventions among people with chronic cardiovascular conditions?

## 2. Materials and Methods

An integrative review of scientific evidence was performed under the recommendations of Whittemore and Knaff [15] who set out 5 successive stages: (1) problem identification, (2) literature search, (3) data evaluation, (4) data analysis, and (5) presenting and interpreting results. The research question was formulated through the PICO reports system, which describes: (a) participants, (b) interventions, (c) comparisons, and (d) outcomes, guiding the search and identification of articles in various databases [16].

The search period encompasses January 2016 to June 2020 in the databases: Web of Science (WOS), Scopus, PubMed and Scielo. Keywords were used in English, Spanish and Portuguese according to the DeCS and MeSH. The keywords used in English were: eHealth and Chronic Disease; in Spanish: eSalud and Enfermedades Crónicas; and in Portuguese: eSaúde and Doença Crônica, which were all combined via the Boolean operator “AND”. The following search filters were used: article as document type, publications within the previous 5 years and English, Spanish or Portuguese language. Since most of the articles about eHealth were published between 2016 and 2020, more evidence has been generated about the use of eHealth strategies, especially in chronic diseases, so we decided to state the last five years as a search filter.

Articles were then selected by reading titles and abstracts, following which articles were read through completely. Inclusion criteria were: (1) Participants: quantitative, qualitative and mixed-method studies involving participant aged between 18 and 80 were considered for both sexes, with chronic cardiovascular ailments (diabetes mellitus, arterial hypertension, dyslipidemia, coronary cardiopathy, cerebrovascular disease, peripheral vascular disease; cardiac failure); (2) Interventions: eHealth interventions delivered in any setting; (3) Comparisons: Not applicable; (4) Outcomes: Barriers and facilitators for the implementation of eHealth interventions. The following were excluded: theses, reviews, protocols, book chapters and editorials. The articles that were repeated in different databases were considered one time only. Subsequently, the articles were exhaustively analyzed for their objectives, methodology, results presentation, discussion and conclusions.

A total of 974 articles matching the aforementioned criteria were found in Scopus (*n* = 248), WOS (*n* = 590), PubMed (*n* = 132) and Scielo (*n* = 4), of which 17 met the inclusion criteria: 10 corresponding to the WoS database, 4 to Scopus and 2 to PubMed (Figure 1).

## 3. Results

Out of all articles selected, eight reported qualitative-type studies and six were quantitative. The number of participants in each study ranged from 12 to 4144 people with chronic conditions and health service providers. Participants were adults, between 18 and 80 years old, with conditions of type 1 diabetes mellitus (T1DM) (*n* = 1), type 2 diabetes mellitus (T2DM) (*n* = 3), and arterial hypertension (ATH) (*n* = 1) with no other associated chronic conditions. Articles were also found including patients with acute myocardial infarctions (AMI) and cardiovascular accidents (CVA) (*n* = 1), ATH, dyslipidemia and congestive heart failure (CHF) (*n* = 1), ATH, DM2, CVA and AMI (*n* = 1), along with other studies without specifications of which particular chronic ailments were present (*n* = 4). The remaining articles had professional nurses participating (*n* = 2) or doctors in charge of formulating policies and health service representatives (*n* = 1). In Table 1, the studies selected for integrative review are displayed with the following data: author, publication year, country, objective, study design, participants, chronic condition and eHealth tool type. Since these studies used various methodologies, no quality evaluation was carried out, but an evidence-level identifier was included, according to the Melnyk and Fineout–Overholt Model [18].

Regarding barriers and aids to eHealth intervention implementations (Table 2), out of the 14 articles selected, 10 studies reported user perspectives, 3 gave health professionals’ perspectives and 1 article mentioned institutional-level aids and barriers.

User-level barriers and aids can be categorized as those associated with: (1) user characteristics, (2) user attitudes and beliefs and (3) technology use in healthcare.

Regarding the barriers associated with user characteristics, age, socioeconomic level, educational level, health status and technological literacy level were identified as influencing healthcare technology use. Increased age was reported as connected to lower use and adherence to healthcare technology, arguing that this age cohort is less familiarized with technology [24,27,30,34] and that people with lower socioeconomic and educational levels also showed lower use of technology [24,30,34]. Health status was also reported as a determinant factor, since a highly complex health condition might impede users from adhering to healthcare technology [30]. Health status was also reported as a determining factor, since a more compromised health status impedes users from adhering as best as possible to healthcare technology. The most frequently mentioned barrier in the articles reviewed was technological literacy level [23,24,26,30,31,34], understood as the knowledge and cognitive and instrumental ability to manage new technologies [35], with results indicating that lower tech literacy levels meant lower use and adherence to the IT.

Among the barriers associated with users’ attitudes and beliefs, senior citizens were highlighted as the group presenting the most prejudices about the use of healthcare technology. One example of this was the belief that technology is meant for young people, leading to them feeling left out as an objective population [20,27]. They also perceived that using technology is synonymous with severity of chronic conditions, mentioning that it promotes the sensation of illness [20,25]. Other barriers described include the lack of motivation to participate associated with the lack of perceived benefits [30] and the view that technology is frustrating and hard to use, generating lack of self-confidence in its use [27,31]. Finally, other reasons given by users for nonadherence to technology was the perception that it meant losing face-to-face encounters with healthcare workers, further depersonalizing healthcare [31]; the lack of privacy was a further concern, since online information could be compromised [26] along with the high cost associated with this strategy type, which would further raise healthcare costs [20].

The third user-mentioned barrier type was associated with technology use in healthcare, where they highlighted that apps lacked a design according to each patient’s needs [21,34], many of which were complex and unintuitive [20], along with problems with logins and Internet connections [25,26].

Some of the aids associated with user characteristics included: being young [24,27,30,34], higher socioeconomic and educational levels [24,29,33], good overall health and high technological literacy [23,24,26,30,31,34]. Some of the aids associated with patients’ attitudes and beliefs included training prior to technology use [32], continuous accompaniment from family and friends, as well as constant monitoring by healthcare personnel to clarify problems [20]. They also highlighted the perception that using this strategy allows autonomy for better managing the disease [20,21,26] and constitutes a support to the patients’ learning process [27]. Finally, an aid associated with technology was a simple intuitive app design along with home internet access [26] providing swiftness, access and exact information to users [27].

Regarding health professionals who are in charge of putting these health strategies into practice and supervising them, various barriers were identified, including platform and app access being only by computer, which delays processes, whereas implementation of access by tablets and cell phones would further facilitate work [22]. Distrust in potential benefits of this technology for users’ health [33] and difficulty in technology use in older and less tech-literate professionals also appeared.

Facilitators mentioned by healthcare professionals were that technology diminished emergency admissions and hospitalizations, as well as improving healthcare quality [22]. Additionally, some benefits were identified, especially for senior citizens, vulnerable populations and highly demanding patients [22]. They also mentioned having greater experience with using technology and the Internet, which facilitated its use in healthcare.

Finally, among the barriers and aids described at the institutional level, it was found that technological systems in healthcare stop at the research project stage without advancing to the implementation stage [28]. The lack of available funding was also highlighted, along with lack of awareness about remote monitoring, anxiety over who would be responsible for the data generated, system design and regulatory standards. As facilitators, it was mentioned that growing demand in health services, education and patient empowerment [28] represent opportunities for this healthcare technology strategy to meet said needs and potentiate eHealth strategies to a degree.

## 4. Discussion

eHealth intervention implementation among patients with chronic cardiovascular conditions in clinical practices is still a challenge, making it fundamental to understand the needs of users, professionals and health organizations to optimize its benefits.

According to the articles analyzed in this integrative review, various barriers and aids mentioned by users in healthcare technology implementation were mentioned. These barriers include low user tech literacy [23,24,26,30,31,34], the feeling that these eHealth strategies depersonalize attention due to lack of face-to-face contact with healthcare providers [31], and how technology is not centered on users’ needs [21,34].

A study in Holland [36] showed that people with high digital literacy levels are more likely to adopt electronic health apps, including online personal health records, compared with people with limited medical literacy. Another study established that this type of technology generates distance between patients and professionals, as it replaces in-person relations with virtual relations [37].

Similarly, a study from 2018 [38] indicates that eHealth strategy design and implementation did not contemplate users’ use needs and contexts, trusting only in intrinsic properties of technologies and ignoring patients’ singularity [36]. We can also highlight that there is a high level of satisfaction and acceptance when patients perceive that interventions are relevant for their needs and receive proper support [39,40]. Furthermore, the holistic framework for improving eHealth technologies’ acceptance and impact indicates the importance of contextual factors as key elements for success, including social and economic contexts as fundamentals [41].

Patients’ distrust over their data privacy is notable [26], referring to a sensation of unease about having their sensitive information violated, an apprehension already reported in other review [42]. Another highlighted limiting factor fundamental in healthcare technology use is healthcare quality [42]; however, this was not identified in studies included in this review.

Some of the aids included perceiving technology as a support system [33] allowing patients autonomy and self-management [43], which was favored by pre-implementation education and support from family or neighbors especially regarding eHealth adherence strategies [44]. These points have been described previously [45,46], when establishing that support for integral educational programs that keep the perspectives of patients and healthcare professionals in mind improves healthcare professionals’ future skills as well as promoting person-centered care [47]. Training content, duration and facilitation are important for eHealth intervention effectiveness, along with user motivation to participate or try using [48].

Motivation is recognized as a fundamental factor for technology use and adherence, which was also mentioned in a 2019 review [39] showing that patient response levels aligned directly with intrinsic motivation levels.

The role of social support identified in this review is confirmed by other studies establishing the facilitating role of family members [48], their emotional support and help in understanding information [49].

Regarding sociodemographic factors, the results suggest that these have an important influence on eHealth use, which can act as barriers or aids to their use. Young people with more education and better socioeconomic conditions present the highest adherence to technology use in healthcare. This was also highlighted by a study [50] reporting that older, lower-income and/or lower-education populations were less likely to use eHealth. People living alone with chronic ailments are also less inclined to use eHealth, since they have no family members helping them with difficulties faced during use [50]. Rural residents have less access and fewer opportunities regarding eHealth; however, they also have the highest need for eHealth to bypass medical attention access problems in remote areas [50].

Lack of perceived usefulness stands out as a major barrier for healthcare providers’ adherence to eHealth interventions [33]. Perceived usefulness is also mentioned as an aid, since it can be seen as helpful to users, being seen as a strategy that cuts emergency admissions and ultimately hospitalizations, as well as this technology bringing in patients with more healthcare access problems. A US study [51] found that leadership is crucial for successful implementation of evidence-based practice, along with indicating that leadership operating environments are important, an aid which was not encountered in the present integrative review.

There is little literature addressing barriers and aids at the institutional level. According to this review, the lack of funding stands out as a principal limiting factor for carrying out projects involving healthcare technology [28]. There is a need to continue searching into this area to have evidence of the improvement in the patients’ chronic conditions, a decrease in complications, mortality, and hospitalizations, and an increase in the patients’ and providers’ satisfaction [7]. What it is expected is a higher allocation of resources to sanitary services using eHealth strategies.

This context was found in other studies [52,53] where the cost category certainly emerges as an eHealth intervention barrier. Observing the distribution frequency of inter-entity category articles, costs were mentioned principally by the health system. In fact, they were mentioned as the most important for both the success and failure of interventions [52]. Infrastructure problems have also been mentioned as an important studied barrier [47]. One principal institutional facilitator is the increasing user demand in healthcare, opening a door to technology as a good strategy for satisfying these needs.

One strength of this review is the incorporation of articles including diverse stakeholder groups and using diverse methods. The COVID-19 pandemic has placed demands on the health systems to implement eHealth interventions more massively, so this information will add to the knowledge on how to maximize their potential. A limitation is that considering articles published in the last 5 years could have limited inclusion of studies with an institutional focus, which would not allow full elucidation of everything occurring at this level and ultimately visualize improvement opportunities. Finally, this review was carried out until June 2020; therefore, it corresponds to a pre-COVID literature review; this also corresponds to a limitation of this study. We suggest carrying out a new analysis with data collected during and after the pandemic period.

## 5. Conclusions

According to this integrative review, the principal barriers to eHealth implementation include low user tech literacy, lack of adaptation of strategies to user needs and depersonalization of care, along with the lack of privacy and connection problems during use. The principal aids include visualizing technology as a patient support system, mentioning the educational component and family support as fundamental for greater eHealth strategy adherence. There is no difference between patients’ barriers and aids in the function of the chronic cardiovascular condition from which they suffer. For professional and institutional aids, the main factor is decreased patient admissions to emergency rooms. Finally, the most mentioned barriers in this area were the lack of perceived usefulness and difficult access to eHealth platforms, along with the lack of financial resources to carry out these healthcare strategies.

## Figures and Tables

**Figure 1 ijerph-19-08296-f001:**
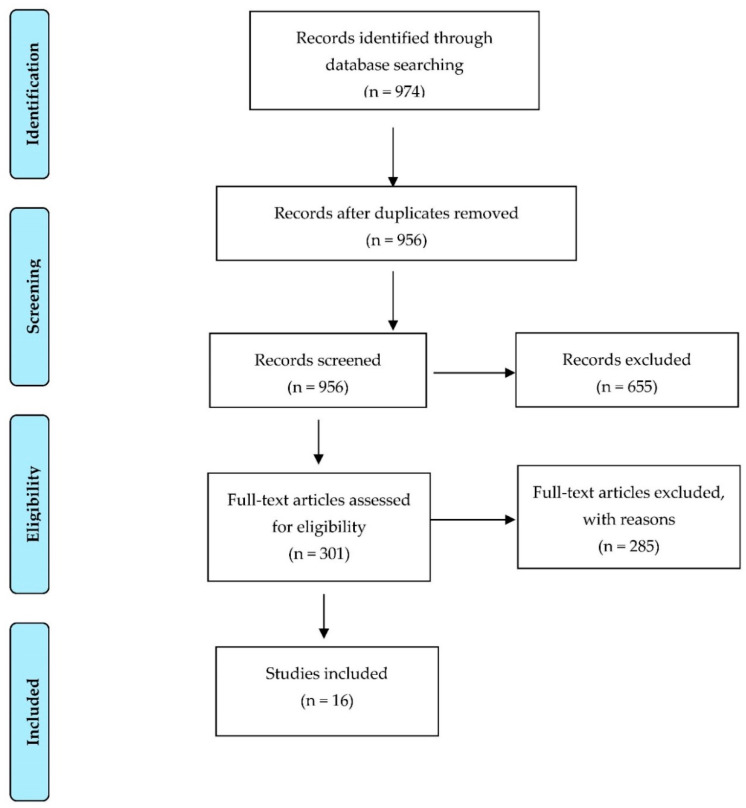
Flowchart for study identification, selection and inclusion. Adapted with permission from: Moher D, Liberati A, Tetzlaff J, Altman D. The PRISMA group Preferred Reporting Items for Systematic Reviews and Meta-Analyses: The PRISMA Statement, 2009 [17].

**Table 1 ijerph-19-08296-t001:** Characterization of Selected Articles.

Author and Year	Country	Objective	Study Design	Participants	Chronic Condition	eHealth Tool	Evidence Level
Steinman et al., 2020 [19].	Cambodia	To understand facilitators and barriers to chronic disease management and the acceptability, appropriateness, and feasibility of mHealth to support chronic disease management and strengthen community-clinical linkages to existing services.	Qualitative exploratory study	70 patients	ATH and DM2	Text messages	Level 6
Hermann et al., 2020 [20].	Germany	Identifying factors supporting adhesion to digital focuses and wider patient acceptance.	Qualitative with grounded theory focus.	20 patients with anticoagulant therapy, age > 65 years.	CVA and AMI	Telemonitoring via smartphone	Level 6
Marsh et al., 2020 [21].	USA	Exploring portal knowledge among emerging adult patients, with perceived barriers and aids.	Qualitative study	27 emerging adults with Type 1 Diabetes Mellitus.	DM1	Virtual health education platform.	Level 6
Gjestsen et al., 2020 [22].	Norway	Identify managers’ and professionals’ perspectives on eHealth intervention use.	Qualitative with case-study focus.	17 health functionaries.	-----------	Virtual health platform.	Level 6
Zigdom et al., 2020 [23].	Israel	Evaluate attitudes among general hospital nurses towards online medical information searching	Cross-sectional quantitative study.	121 nurses at 3 general hospitals.	-----------	Online database information search.	Level 6
Ernsting et al., 2019 [24].	Germany	Identify factors associated with chronic patients’ mobile health app use.	Quantitative study of secondary data analysis.	1500 participants in a Web-based survey.	ATH, DM2, CVA, AMI	Pfizer Monitor mobile application	Level 6
Jeffrey et al., 2019 [25].	Australia	Evaluate experiences, barriers and aids to app use among people with Type 2 diabetes.	Qualitative	30 people with DM2 diagnoses.	DM2	Mobile health application.	Level 6
Mangin et al., 2019 [26].	Canada	Examine attitudes and online eHealth record use in patients with chronic cardiovascular disorders	Cross-sectional quantitative study.	693 patients with primary care chronic cardiovascular conditions.	Chronic unspecified cardiovascular conditions.	Virtual health platform.	Level 6
Milos et al., 2019 [27].	Sweden	Exploring senior citizens’ attitudes and beliefs to better understand the factors influencing eHealth adhesion.	Qualitative study	15 primary care left patients with chronic conditions	Nonspecified chronic cardiovascular conditions.	Mobile health application.	Level 6
Díaz et al., 2019 [28].	Ireland	Exploring barriers and aids to institutional-level eHealth technology adoption.	Summary of consensus at eHealth Innovations for Home and Community Care conference.	Doctors, research professors, policymakers and health service representatives.	-------------	Cardiac telemonitoring.	Level 7
Rhoads et al., 2017 [29]	USA	To identify the potential factors that influenced the use of m-health technology and adherence to the control of hypertension symptoms	Nonrandomized controlled study	48 women	ATH	Telemonitoring of arterialpressure	Level 6
Ernsting et al., 2017 [30].	Germany	Exploring reach of smartphone and health app use and their use behavior in chronic patients.	Quantitative correlational study.	Survey of 4144 patients over 35 years old with chronic cardiovascular conditions.	Unspecified chronic conditions.	Mobile health app.	Level 6
Stangeland et al., 2017 [31].	Norway	Exploring experiences with a GSD-based eHealth intervention and understanding reasons for leaving it.	Qualitative study.	12 adults with DM2 who left an eHealth intervention.	DM2	Mobile health app.	Level 6
Ondiege et al., 2017 [32].	England	Exploring hypertension patients’ beliefs and worries about monitoring devices	Qualitative study	20 cohabitating couples suffering from hypertension	ATH	Arterial pressure telemonitoring.	Level 6
Duplaga et al., 2017 [33].	USA	Evaluating skills, technology use and exploring opinions about the health area among nurses.	Quantitative cross-sectional study.	628 nurses took a questionnaire to evaluate technology use in healthcare	-------------	Virtual health platform.	Level 6
Granger et al., 2016 [34].	Australia	Determining whether greater smartphone-Tablet familiarity was associated with higher eHealth use,	Quantitative correlational study	1865 participants with cardiovascular conditions who used technology.	Unspecified chronic conditions.	Mobile health app.	Level 6

**Table 2 ijerph-19-08296-t002:** Barriers and aids to eHealth interventions for users, healthcare professionals and healthcare institutions.

Article	Level	Barriers	Aids
Steinman et al., 2020 [19].	Patient	- Limited time and resources to access to pharmacological treatment, clinical support and recommendations of physical activity and healthy diets. - Lack of equity in the access to quality and effective chronic care. - Living in rural areas - Limitations of technology literacy	- The system is seen as an opportunity to remember prescription, clinical analysis and medical consultancy - Education on the best practices for the management of chronic diseases at home. - Support for those barriers that cannot be easily overcome. - Use of voice messages over text messages for familiarity.
Hermann M et al., 2020 [20].	Patient	- Perception that constant health monitoring promotes feeling of illness. - Sensation that technology is only for young people. - Including technology in health only increases costs.	- Technology use gives users feelings of autonomy in health management. - Support from family or neighbors facilitates use.
Marsh K et al., 2020 [21].	Patient	- User perception that information portal is extremely general and not lefted on particular user needs. - Other, easier information access modes like the Internet exist apart from the portal.	- Sensation that the portal facilitates providers’ access to medical history, supporting diabetes self-management
Gjestsen M et al., 2020 [22].	Healthcare Professional	- Platform access is only from computers, without considering alternatives including tablets or smartphones.	- Healthcare professionals say that it cuts ER intakes and improves home care quality. - Perception that virtual platform mainly benefits senior citizens.
Zigdom A et al., 2020 [23].	Healthcare Professional	- Low tech literacy. - Greater age equals less Internet and tech use.	- Social media experience generates better attitudes towards Internet use for finding healthcare information.
Ernsting C et al., 2019 [24].	Patient	- Little or no tech literacy.	- Younger people use the app more. - More educated people use tech more.
Jeffrey B et al., 2019 [25].	Patient	- Participant lacked knowledge about apps as healthcare tools. - Perception that app use equals more severe illness. - Bad rural connections.	- Visual trend representation, intuitive navigation and convenience (e.g., discretion and portability).
Mangin D et al., 2019 [26].	Patient	- Disuse of technology. - Sensation of lack of privacy. - Internet connection loss.	- Home internet access. - Perception that tech use allows for health self-evaluation, and therefore autonomy.
Milos V et al., 2019 [27].	Patient	- Incuriosity among seniors about tech use. - Sensation of lack of self-confidence in tech use.	- Desire to join due to need for information and learning support. - Sensation that tech brings potential health advantages. - Users feeling need for speed, access and correct integral information.
Díaz Y et al., 2019 [28].	Institutional	- Health tech systems stall in project study stages. - Lack of funding - Lack of awareness about remote monitoring. - Anxiety about responsibility for data generated.	- Growing demand in patient services, education and empowerment.
Rhoads et al., 2017 [29].	Patient	- The use of the system compromises the privacy of the user when sending the data to the reference health left. - Using the system daily demands too much time - The use of technology generates feelings of anxiety, and discomfort since the measurement values may be not precise enough.	- To have the knowledge to use the system at home - Help from close persons to solve mHealth difficulties at home. - Feeling that the system warns of risks and avoids complications - Easy to use system, clear and understandable. - Perceived satisfaction
Ernsting C et al., 2017 [30].	Patient	- Wealthier people adhere more to tech use - Poorer people adhere less to tech use - Lower literacy equals lower chance of using Smart devices and mobile health apps.	- Younger patients are likelier to use smart devices and mobile health apps. - The better the health status, the more likely patients will use smart devices and mobile health apps.
Stangeland S et al., 2017 [31].	Patient	- Lack of eHealth intervention participation motivation. - Sensation that tech is frustrating. - Perception that content is irrelevant and incomprehensible. - User preference for doing other activities. - Perception that tech loses face-to-face contact.	- Category not included.
Ondiege B et al., 2017 [32].	Patient	- Batteries in equipment had very short life. - Difficulties handling device. - Pressure equipment had low sensitivity.	- Sensation of useful telemonitoring. - Training prior to program start.
Duplaga M et al., 2017 [33].	Healthcare professionals	- Distrust among nurses regarding real contributions of tech apps.	- Internet use experience increases professionals’ trust in health apps.
Granger D et al., 2016 [34].	Patient	- The lower the familiarity with tablets or smartphones, the lower the chance of using this type of tech.	- Better chances of adhering to health interventions when adapted to the needs of each user. - Younger and more educated people tended to use technology in healthcare more.

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
