# Peer review of "Barriers and Supports in eHealth Implementation among People with Chronic Cardiovascular Ailments: Integrative Review"

_ijerph, 2022, doi:10.3390/ijerph19148296_

Round 1

Reviewer 1 Report

Introduction

This section is poorly written and unable to convey the rationale of conducting this review. It should be re-written or help from a native English speaker should be sought to make it readable.

Material and methods

Authors selected PICO framework to formulate their review question, but have not stated the question. The selection of PICO framework and search strategy are not consistent with each other and the eligibility criteria.

Search strategy is not comprehensive, especially not searching PubMed or Medline means that significant amount of relevant literature might not be retrieved

Rationale for the eligibility criteria is missing. For example, why only articles published in past five years were included in the review?

Screening process is not described, hence lacked transparency and potential for reproducibility

PRISMA guidelines were not mentioned and used. Hence, reporting of systematic review results is not consistent with recommended reporting guidelines, which makes this review less reliable. 

Author Response

Dear Reviewer

Please find attached the revision of manuscript ID1719150 entitled "Barriers and upports in ehealth implementation among people with chronic cardiovascular ailments: integrative review" We very much appreciate your invitation to revise the manucripst.

We would like thank the reviewers for their thoughtful review of the manuscript. They raise important issues, and their inputs are extremely helpful for improving the document. We agree with almost all their comments, and the entire paper has been revised in the line with these suggestions.

Our response to the reviewers' reports is provided below, All Changes in the new version of the manuscript have been highlighted in yellow.

Thank you very much for yor assistance and guidance.

The authors

Reviewer 2 Report

In this manuscript titled, " Barriers and upports in ehealth implementation among people with chronic cardiovascular ailments: integrative review.", Sophia Herrera et al., authors focused on identifying barriers and aids to eHealth intervention implementation in people with chronic cardiovascular conditions. This manuscript is written clearly, however, the manuscript appears preliminary.

  1. In this study, the authors searched the publications within the previous 5 years and pertinence to the research objective. Why selected only 5 years, not 10 or 20 years?
  2. In the conclusions, the authors suggested that the most mentioned barriers in this area were the lack of perceived usefulness and difficult access to eHealth platforms, along with the lack of financial resources to carry out these healthcare strategies. So how improve this situation?

Author Response

Dear Reviewer,

PLease find attached the revision of our manuscript ID 1719150 entiled "bariiers and upports in ehealth implementation among people with chronic cardiovascular ailments: integrative review". We very much appreciate your invitation to revise the manuscript.

We would like to thank reviewers for their thoughtful reviw of the manuscript. They raise important issees, and their inputs are extremely helpful for improving the document. We agree with almost all their comments, and the entire paper has been revised in line with these suggestions.

Our response t the reviewers' reports is provided below. All changes in the new version of the manuscript have been highlighted in yellow.

Thank you very much for oyur assistance and guidance.

The authors

Reviewer 3 Report

An important, popular topic is discussed in this manuscript.

Row 99: How was the median found? In the Table 1 there isn’t listed study enrolling 698 patients and taking the mean of

Row 101-108: “n” refers to the number of articles?

Materials and methods section: a flow-chart/diagram of the study/article selection method should be included.

References: numbering should revise.

General comments:

- the review focused at patients with chronic cardiovascular conditions, but among the barriers and aids, listed in Table 2, no specific item/element/condition characteristic to this patient-group is mentioned. Based on this observation more research articles could be included in this study, if only the existence of a “chronic disease” would have been the inclusion criteria.

- this topic is an important one, which literature has grown very fast in the last two, COVID-19-dominated, years. In PubMed, searching after keywords as, eHealth/telemedicine AND chronic diseases AND cardiac/cardiovascular applying the filter publication date at 1 year more than 40 publications are shown. In this context, a short overview of the new information regarding this topic should be included in the manuscript, in this area information from 2020 could be partly out of date.

Author Response

Dear Reviewer:

Please find attached revision od manuscript ID1719150 entitled "Barriers and upports in ehealth implementation among people with chronic cardiovasculas ailments: integrative review". We very much appreciate your invitation to revise the manuscript.

We would like thank the reviewers for their thoughtful review of manuscript. They raise important issues, and their inputs are extremely helpful for improving the document. We agree with almost al  their comments, and the entire paper has been revised in line with these suggestions.

Our response to the reviewers' reports is provided below. All changes in the new version of the manuscript have been highlighted in yellow.

Thank you very much for your assistance and guidance.

The authors

Reviewer 4 Report

Dear editors,

Thanks for the opportunity to review this article, which describes an integrative review about barriers and facilitators in eHealth implementation among people with chronic cardiovascular ailments. I found this article interesting. Nevertheless, some aspects could perhaps be revised to help readers better understand the authors' results and conclusions. These are the following:

The authors described the searching process in methods, but this description does not seem accurate. Scientific research must be reproducible. Thus, I would invite the authors to improve this section, offering additional important aspects. For example, the authors describe that searches were performed between May and June 2020, and one of the search filters applied was publications within the previous five years. Thus, the readers do not precisely know the limits of the searches performed, and the number of articles could vary if they are completed almost two months before or after. Another important aspect here is that the authors also describe «pertinence to research objective» as a search filter. But this is not a search filter; this is a filter that perhaps should be applied when they have obtained the results of the searches, as you can not set this filter in the databases consulted. Another important aspect is that the authors should better describe the criteria used to eliminate the discarded articles. Thus, I would invite the authors to rewrite this section and add a flowchart.

In the limitations section, I think other limitations could be described. More potential selection biases can be added to the time limitation of the searchers. If the 5-year period limits the research, I will invite the authors to increase this period, as this is an easy-to-solve limitation. Other potential limitations related to the methods must be better described, or the qualitative nature of many of the articles retrieved. I would invite the authors to assess them

Author Response

Dear Reviewer,

Please find attached the revision of our manuscript ID 1719150 entitled "Barriers and upports in ehealth implementation among people with chronic cardiovascular ailments: integrative review". We very much appreciate your invitation to revise the manuscript.

We would like to thank the reviewers for their thoughtful review of the manuscript. They raise important issues, and their inputs are extremely helpful for improving the document. We agree with almost all their comments, and the entire paper has been revised in line with these suggestions.

Our response to the reviewers' reports is provided below. All changes in the new version of the manuscript have been highlighted in yellow.

Thank you very much for your assitance and guidance.

The authors

Round 2

Reviewer 1 Report

Thanks for talking out time in reading comments and trying to respond them. 

I think that authors have not addressed my comments adequately. 

For example, in my comment I said that introduction section does not convey the rationale for conducting this review, while authors' mentioned  'some wording adjustments to make the writing appropriate'.

In addition, authors have cited the PICO framework, but not addressed my comment of 'PICO framework and search strategy are not consistent with each other and the eligibility criteria'.

Also reading of question does not suggest it to be following a PICO framework, e.g., there is no comparator in the question. 

Author Response

Dear Revisor,

Our response to your reports is provided below:

Comment 1: For example, in my comment I said that introduction section does not convey the rationale for conducting this review, while authors' mentioned ' some wording adjustments to make the writing appropriate'.

Response 1: Sorry for not addressing completely the issue. We add in the Introduction the text below:

Interventions in e-health are expected to continue to increase, as they are a cost-effective option, in the face of different limitations of traditional interventions, such as mobility restrictions, for example, those derived from the covid-19 pandemic. Thus, knowing the experiences in the implementation of eHealth initiatives will help in their future development. This study addressed chronic cardiovascular diseases due to their high prevalence worldwide. A review of E-health interventions is necessary due to the rapid increase of knowledge on the subject, moreover, reviews in recent years have focused on interventions aimed at specific chronic conditions such as allergies [10] or depression [11], or included a variety of chronic conditions such as diabetes, HIV, pain, and epilepsy among others [12]. Other reviews have focused exclusively onthe children or adolescent groups [11] or the older adult population [13]. Reviews of reviews on the subject have been published more than 10 years ago [14].

Comment 2: In addition, authors have cited the PICO framework, but not addressed my comment of 'PICO framework and search strategy are not consistent with each other and the eligibility criteria'.

Response 2: In the new version, we included eligibility criteria according to the PICO framework.

Comment 3: Also reading of question does not suggest it to be following a PICO framework, e.g., there is no comparator in the question.

Response 3: The question that guides this review is posted according to the PICO framework since it contains the (P) Patient (I) Intervention and (O) Outcomes; however, the (C): Comparison is not included. The reason is that according to the literature review, it is highlighted that sometimes there is no intervention to compare with, as in the current review.

The following paper and academic website that mentions it:

Martinez J, Ortega V, Muñoz F. [Design of clinical questions in evidence-based practice. Formulation models]. Enferm glob [Internet]. 2016 [cited 2022]; 15 (43). Available from: https://scielo.isciii.es/scielo.php?script=sci_arttext&pid=S1695-61412016000300016

University of Canberra Library. Evidence-Based Practice in Health. 2022

https://canberra.libguides.com/c.php?g=599346&p=4149722

All changes in the new version of the manuscript have been highlighted in yellow.  

Reviewer 2 Report

 Accept in present form

Author Response

Dear Revisor,

Thank you for your careful review of the manuscript.